# The Impacts of COVID-19 on Technological and Polytechnic University Students in Mexico

Lourdes Vital López [1], Marco Antonio Zamora-Antuñano [2], Miguel Angel Cruz-Pérez [2], Juvenal Rodríguez Reséndíz [3], Fabiola Fuentes Ramírez [1], Wilfrido Jacobo Paredes García [3], Hugo Rodríguez Reséndiz [3], Marisela Cruz Ramírez [4] and Raul García García [4,*]

1   Carrera de Mantenimiento Industrial, Universidad Tecnológica de Tamaulipas Norte, Universidad Tecnológica, Reynosa 88680, Mexico; lourdes.vital@uttn.mx (L.V.L.); fabiola.fuentes@uttn.mx (F.F.R.)
2   Posgrado y Centro de Investigación, Innovación y Desarrollo Tecnológico de UVM (CIIDETEC-UVM), Universidad del Valle de México, Queretaro 76230, Mexico; marco.zamora@uvmnet.edu (M.A.Z.-A.); miguel_cruzp@my.uvm.edu.mx (M.A.C.-P.)
3   Facultad de Ingeniería, Universidad Autónoma de Querétaro, Queretaro 76010, Mexico; juvenal@uaq.edu.mx (J.R.R.); wparedes17@alumnos.uaq.mx (W.J.P.G.); hugorore@uaq.mx (H.R.R.)
4   División de Química y Energías Renovables, Universidad Tecnológica de San Juan del Río, Queretaro 76800, Mexico; mcruzr@utsjr.edu.mx
*   Correspondence: rgarciag@utsjr.edu.mx

**Abstract:** The contingency measures put in place by the government during COVID-19 exposed the students to a new condition to which they must adapt. To understand how the students perceive and cope during the unplanned, changed learning mode, we conducted a study using an evaluation tool which seeks to understand the effect of the contingency measures associated with the emergence of the COVID-19 virus on the students. By assigning a data collection instrument to students who are part of 15 technological universities (TUs) and 7 polytechnic universities (PUs), we determined how they were affected by COVID-19. The questions intended to evaluate the social, economic, academic, emotional, and health effects experienced. A total of 6596 students were assessed in the study representing an appropriate percentage of the Mexican students. The outcome of the study showed that 12% of the students agreed with the online approach to learn adopted because of the contingency. A total of 39% reported that they have a good environmental space for online learning, 32% reported that they mostly take their classes via their mobile phones, and 3% said they lacked access to an internet facility and as such could not take their classes. A total of 14% reported that they have little access to the internet, while 42% reported that they regularly have internet, and both complained that internet fluctuation significantly affects their academic performance. Comparing the different modes of teaching, 52% believe an in-person class is the best approach to learning, but 22% agreed that a hybrid system will be effective. Through a multiple correspondence analysis (MCA) it was determined that, in the effects, there was no significant difference in relation to gender. The effects that most impacted the students were economic, connectivity, and a lack of physical activity.

**Keywords:** COVID-19; online education; student opinion

## 1. Introduction

The COVID-19 pandemic originating in Wuhan, China [1], has had an impact on education worldwide [2]. The impromptu emergence of this virus enforces a need to change the mode of educating students, resulting in the adoption of a technology-dependent mode of learning [3], with interactive virtual campuses being an immediate future objective in education [4]. No one was prepared to migrate from a face-to-face educational model to a virtual one without question. The World Health Organization (WHO) stated that COVID-19 went from an epidemic to a pandemic on 11 March 2020 [5]. Therefore, in Mexico, in-person

classes were suspended for all educational levels on 20 March 2020 (Civiles, 2021). In response, the education sector was forced to change to online and virtual classes using different platforms, such as MS Teams, zoom, meet, and Canvas, among others [6], which were already used but as a way to increase the ways in which students could interact among them and with the information. The pandemic changed this state and made those platforms the only way to learn academic content. These technological tools have been used for more than a year. Some authors highlighted the difference in the scenario in the face of the pandemic and online teaching, denominating it as the "First Emergency Remote Teaching" [2,7,8].

Rodríguez-Segura (2020) analyzed the academic environment for traditional changes to remote learning using the MicroSoft Teams platform with a satisfactory approach considering knowledge assessment and skills achievement [6]. The main results established significant percentages, where more than 60% of the students surveyed demonstrated their satisfaction in the use of the Teams platform and the organization of class sessions and the activities developed by teachers. The pandemic, in the education sector, affected not only the teaching methodology but also the continuous evaluation tests and global exams or assessments with a physical presence of students in classrooms and/or laboratories. Videoconferencing classes offer advantages, drawbacks, and tips. However, the criteria for online evaluation must be clearly explained and should be a way to develop the digital literacy requirements for a lockdown. This type of evaluation suggests using a plagiarism analysis tool for task reviews, requesting work in teams or in pairs, making presentations individually or in groups, applying individual oral tests using videoconferencing tools, and greater flexibility to ensure the inclusion of students, among others [9]. The design of quality online activities should have general principles, such as going from easy to difficult, variability, clear standards, rational temporal organization, specifying evaluation criteria, and offering individual and group activities [10], as a way to learn during the lockdown but also as a new way to socialize during class time. This could have been a great opportunity for digital literacy training and to reduce the digital gap. On the other hand, the effect of online or remote learning on medical education during the pandemic on medical residents, surgical specialties, and those that base their learning on procedures [11], such as in dentistry, were different. Duran-Ojeda (2020) noted that academics could ensure theoretical progress during the pandemic, but the progression of practical activities was postponed until, in each country, the quarantine measures were halted and education returned to normal [12], which made students feel that the focus of their program was not being delivered. COVID-19 will have a long-lasting impact on the environmental health field and will open new research perspectives and policy needs [13] because of the generated isolation, the lack of procedural learning, and the impact on emotions stemming from the situation. This lack of practice generated a perception of wasting time or a reduced quality in the program for the students of programs with a high percentage of practical activities or laboratories.

In another study conducted by Son (2020), through a survey of 195 university students, the results revealed that 71% of the students reported an increase in stress and anxiety due to the COVID-19 outbreak [14]. Also, multiple stressors were identified that contributed to increased levels of stress, anxiety, and depressive thoughts among students [14]. Therefore, it will be necessary to innovate and implement an alternative educational system and evaluation strategies [15] as the COVID-19 pandemic has generated surprising and unexpected experiences among college students [16] which were not as common in face-to-face schooling. Undoubtedly, the precautionary and preventive measures taken to curtail this pandemic, including online learning, have affected the social and educational aspects of students' lives [17]. Thus, the attitude toward distance learning during confinement has been evaluated [18], as well as a variety of psychological impacts that COVID-19 has on students [19] and, more precisely, on students learning practical and procedural skills. There are still many questions to be resolved, for example, to understand the effects caused by COVID-19 on students of the technological (TUs) and polytechnic (PUs) universities of

Mexico, evaluating the situation from March 2020 to June 2021, which have undergone the forced change to virtual-mode teaching during the pandemic [20].

The General Coordination of Technological and Polytechnic Universities (GCTPU) is a part of the Ministry of Education in Mexico (SEP) as part of the secretariat or higher education. The coordination works as a scheme of higher education to fulfill the requirements of the society for students to be integrated into the productive sector with a committed and consolidated teaching team. This system seeks national and international recognition for its efficiency, effectiveness, relevance, and equity, and it is linked to the social and productive sectors that contribute to the economic development of the country [21]. The academic offerings are equivalent to the community colleges of the higher technical universities in the United States and Canada and to the university institutes of technology (IUT) in France [22]. These are university systems that prepare technicians to be able to immediately enter the labor market or to continue with higher education. The educational model has a specific formation scheme: 30% theory-based and 70% practice-based. This model was established in Mexico in 2001 to address professional and qualification needs. It allows for future technicians and engineers with a higher level of education to enter the productive sector with professional internships. In 2021, there were 114 technological universities and 62 polytechnic universities in Mexico. The GCTPU serves 300,000 students, which represents 7.5% of the total number of students (4 million) in higher education in Mexico [23]. Other studies [24] have had different perspectives on the virtualization of the teaching–learning process during the pandemic. Gender and age seem to be important factors in students' satisfaction, so it is of interest to find out if that behavior repeats in the specific model of the GCTPU while using the virtual learning platforms (VLPs) recommended by the institutions. This study is important for reinforcing the public HEIs research on these matters because most of the studies of this kind have been about private HEIs. Modgil (2019) explored how adjunct higher education faculty perceive using social media (SM) as an instructional tool for their students during the pandemic and found out that it is one of the most effective tools for knowledge dissemination [25]. In other related studies, they evaluated the reliability and effectiveness of different information technology (IT) tools for knowledge transfer and found that most of it was based on learning tools (LT), the use of mobile devices (such as cell phones and tablets), and the virtual library (VL), with 89%, 85%, and 82%, respectively [26]. In addition, they found that the use of cell phones made the teaching–learning process more dynamic [27], but it also needs to be studied during emergency remote teaching and particularly in TUs and PUs: now it is not something to introduce variety during class but something that becomes the ordinary and necessary device to attend the class. Adopting an online-learning approach has proven to be the alternative to a physical classroom in an uncontrollable situation. This has allowed and forced universities, faculties, and students to have patience and resilience, both useful ways to face future challenges in high-quality education [28–30] for future professionals. But how can this be studied and incorporated into the curricula? Another effect of the pandemic is a quality diminishment in the expected educational level as explained before regarding the practical and procedural contents. The absence of an online learning infrastructure could have worsened the situation worldwide [31]. The physical distancing amid the pandemic has influenced the attitudes of the teachers worldwide opting for social media (SM) use in online learning, mainly in developing countries; switching to online learning using SM under challenging situations like the COVID-19 pandemic is thus inevitable [32,33]. This contrasts with the use SM had for academic purposes before the lockdowns and the sense of urgency the situation created for digital literacy. SM was a way to enrich the class with different interactive ways to search for data and resources to discuss during class, but during the isolation, it is almost the only way to interact with others, and personal interaction was the desired but avoided way to enrich the class. Many students have lost close relatives and must continue to study under these conditions. Our main interest is not all the universities in Mexico: HEIs are classified into six large groups—public universities, technological education, technological and polytechnic universities, private institutions,

normal education, and other public institutions [34]. All of them switched to a virtual mode of teaching during the pandemic. In this study, we explore the COVID-19 effects on the social, economic, academic, emotion, and health of the students of the TUs and PUs of Mexico during the period March–August, 2021 [23].

*1.1. Theoretical Framework*

University life becomes a factor considered essential for developing professionals: it exerts influence on lifestyle. If it is not adequate, health and good academic performance can be affected [35]. In the face of the pandemic, new educational methods were generated at all levels. Given this, social networks played a key role, influencing a change in the learning scheme that can generate stress and/or anxiety. Even before the lockdowns, these were important concerns for the TUs and PUs. Learning based on social networks is asynchronous, allowing students to interact at their own pace with the educational content at any time or place [36]. However, this causes a change in routines and uncertainty before the development of their education in the future, which can generate different levels of stress, and finally anxiety. Passing through university is a fundamental stage, as it directly influences the professional future, and the social, economic, and psychological changes faced define the development of university students [35,37]. New demands, competitiveness, and economic crises which are not always attended can lead to anxiety [38]. Environmental stressors generate alarming figures of academic stress in students, which causes various physical and/or psychological reactions that must be addressed in a timely manner [39]. If an isolation situation is added to these situations, the complexity will have an impact on emotional and academic performance.

Academic expectations, both internal and external, can be a source of stress. The foregoing leads to the proposal of efficient and appropriate intervention strategies that contribute to the understanding of the sources of stress, such as the academic environment [39]. Anxiety in students can be triggered by external situations or by internal stimuli that may cause physiological and behavioral reactions [40] not always detected and treated by professionals. Alongside these factors, the new strategies forced by the pandemic to continue school activities were based mainly on the intensive use of the internet. However, there are regions that did not have the necessary infrastructure for a rapid implementation of these strategies and emotional tutoring was not robust in the educational system. The use of the internet by university students in developing countries has an effect on academic performance and communication [41]. Not having the necessary technology to achieve educational objectives can generate stress and anxiety. Gladly, there are easy-to-implement techniques with formative assessments to make students feel comfortable, serving as a mediating effect between anxiety and educational performance [39], although the personal or family stressors are not always considered.

Anxiety can be considered as a normal adaptation response to what is considered a threat (stress). It allows a person to improve their performance, even when sometimes this response may not be adequate. Stress can be excessive for the resources available to the individual though [42]. Faced with these abnormal situations, whether temporary or not, coping strategies must be adopted. These strategies are responses that human beings give to solve various events or situations with the resources available to the individual that will be decisive for the effectiveness of such coping [42]. Coping can have two functions: "the regulation of stressful emotions and the modification of the problematic relationship between the person and the environment that causes stress". Regulation is oriented to what is thought and done in situations that generate stress; it is contextual and may be modified as the encounters take place, but it will be influenced by the personal assessment regarding a real demand of what is faced and with what resources it will be addressed [42]. The stressors and affectations should be known to be able to introduce changes in academic environments that lead to changes in society.

### 1.2. Gender Impact during COVID-19

In another study, Vital (2022) described the influences of gender [21] as "an important issue, as a trending topic and more during the pandemic. There are important advances in this regard throughout recent years such as technology improvements, but there are some challenges to face, such as gender equity in access, in digital devices ownership, in training for digital fluency, and ability accessing technology". Although affordability of technological devices is a key factor for exclusion, an analysis should be conducted in specific working environments and working opportunities as a financial inclusion factor in the TUs and PUs [43,44]. Technology represents an omnipresent element that affects globally, and the internet assumes incorporating the individual to an interconnected society where inclusion represents a competitive advantage in development, integration, and wellness [9,45]. The imbalanced coverage in connectivity and technology appropriation generate a digital gap between those with access and those without coverage. This gap could be attributed to a geographic, economic, cultural, and generational disparity. Alva (2015) declared the presence of a digital gap in three dimensions: access, use, and appropriation. Alva explained that these dimensions give three particular gaps: (a) a digital gap of use, (b) an age-range digital gap between native and digital immigrants, and (c) a gender digital gap [46]. The digital gap could be attributed to diverse factors and, according to the Instituto Nacional de Estadística, Geografía e Informática de México (INEGI), the factors are school attendance; being 15–17 years of age; being predominantly male; education level, where the years attending school for a female 15+ is lower than 15+ males; a lower participation of females (23.7%) in Science, Technology, Engineering and Mathematics (STEM); cultural matters as reading habits, which is lower in females 25+; a low attendance of females at cultural activities (39.8%) promoting their personal development; and a lower economic participation by sector in the country which is of 95 per 100 males in the population in the age range 30–49. These factors maintain a rough affectation, which stood up during the COVID-19 pandemic: 66.6% of females 12+ work 30.8 h performing nonpaid activities, their need to work, and being in charge of the family and housekeeping, or by being pregnant, which makes their activity a nonpaid one, while males use 28% of their time, or 11.6 h, for those activities [47]. Korlat (2021) studied four components of digital learning that are susceptible to the stereotyped gender gap [48]. While Lawal et al. (2021) did not find significant differences regarding gender, it is attributed to the fact that both genders were submitted to similar COVID-19 protocols during the pandemic [49]. Females experience unique health risks resulting from their gender. Many of these studies identified inequity in the academic world for females [50]. The barriers include disparity in economic compensation and inequity in the three pillars of academic assessment: teaching, service, and research. Regarding the scientific literature produced before the pandemic, Tkacová (2022) considers that teacher exposition to psychosocial risks derived from the school environment and activities is also a future possibility for distance teaching. This implies a greater chance of depression, stress, and mental health issues [51]. The literature also points out the relationship between inadequate working conditions and psychosocial consequences, such as stress, dysphonia, and voice-related problems, physical inactivity during free time, and anxiety. This reality could also be different in the labor market according to gender. From these differences, the literature centers its attention in the higher exposition of females to domestic violence during lockdowns, and in the working environment, it is legitimate to consider that females could be overwhelmed [52].

### 1.3. Objectives

The main objective, already stated in the introduction, has the following specific objectives:

- To determine if there were significant differences according to gender in the physical, emotional, and health affectations derived from COVID-19 in the TU and PU students.
- To identify the main factors that affected the academic performance of the TU and PU students during the pandemic.

*1.4. Hypothesis*

**Hypothesis 1 (H1).** *There are significant differences in the physical health and emotional state of the TU and PU students in relation to gender during the confinement derived from COVID-19.*

**Hypothesis 2 (H2).** *There are no significant differences in the physical health and emotional state of TU and PU students in relation to gender during the confinement derived from COVID-19.*

## 2. Problem Statement

The strong presence of emotional support and guidance during the learning process is consistent with the results provided by other studies, but never as extreme as it was necessary for the isolation. Barragán's concept of online learning mentions the emotional and significant competence in teacher training for learning, an exercise of its tutorial (pastoral care) function in virtual environments with a strong presence during online learning. Very notable emotions in learning experiences through virtual platforms lead to required self-management to defeat or control anxiety. The physical or emotional health problems of students affected by the COVID-19 pandemic have not been sufficiently addressed in Mexico's higher education institutions [53,54] as the content of the subject is considered the main purpose and, in many cases, emotional problems are not considered in the academic environment. In this sense, and despite the evidence mentioned, in the TUs and PUs, there are no studies that relate the problems mentioned in regard to COVID-19 with anxiety or physical and emotional health in university students. For this reason, the main objective of this research is to determine the main problems that have impacted students due to COVID-19 and identify if certain affectations differed according to gender.

## 3. Methodology

In this study, we used a nonexperimental design at an observation level. The required information was obtained at the field level in a single moment using a quantitative, descriptive, analytical/correlational methodology for finite populations. The effects obtained from the questionnaire and analyzed were divided into six effects: sociodemographics, academics, economics, emotional, social, and health. These effects were selected by the commission in charge during the distance education modality due to the mandatory social isolation in the 2020–2021 school year.

This section presents the development of the *"Evaluation of the impact and consequence of health contingency by COVID-19 during the educational process in students of the Technological and Polytechnic Universities of Mexico"* study. Context and environmental information were collected at the beginning of the study during the teaching–learning process that resulted in the impact on five effects: academic, social, economic, emotional, and health. Figure 1 shows the steps for conducting the research.

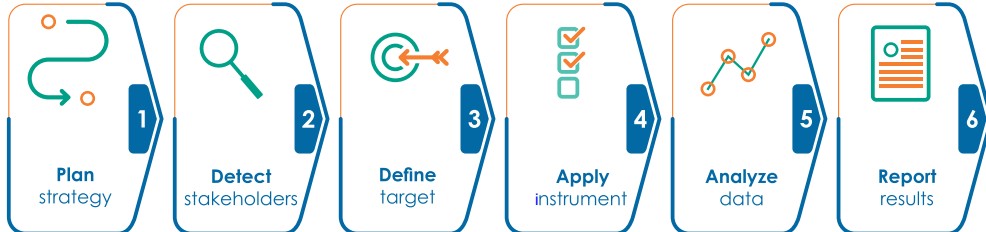

**Figure 1.** Steps for conducting the research.

- Step 1: Field work planning.
- Step 2: Definition of stakeholders and institutions.
- Step 3: Development and validation of data collection instrument.
- Step 4: Application of the data collection instrument.
- Step 5: Data analysis.

- Step 6: Elaboration of the final report and proposal of strategies to improve the educational process.

*3.1. Data Collection Instrument*

Once the objectives were established, an ad hoc questionnaire was developed by 22 teachers (15 from TUs and 7 from PUs) that were working and selected by commission to be in charge of the study. The survey included, in addition to sociodemographic data, aspects related to the confinement. The data collection instrument included 25 questions: 24 items (questions) that described the contextual environment for each category and 1 open-ended question on major problems. It used a Likert scale with a maximum of 5 points. A numeric value was assigned to encode the effects into a statistical system. The participants responded by choosing the value that most fit their feeling and were expected to respond to an assigned open question about the educational strategy that helped them improve their performance in virtual classes [55]. In Appendix A, a table with the data collection instrument presents the analysis of the questions answered by the students at the different participating universities. To validate the data collection instrument, a panel was carried out, resulting in a Cronbach's Alpha of 0.78 in the pilot test; thus, it was considered adequate. It was then delivered to 22 universities so that the participating students could decide to participate or not. Subsequently, the information was stored in a database that quantifies and graphs the total points for the responses for the analysis. Table 1 shows the effects and the number of questions that made up the survey applied to students from TUs and PUs.

**Table 1.** Contents of the data collection instrument.

| Effects | Description |
|---------|-------------|
| General | Contains four questions, including: affiliation , academic program, gender, and age of students (Q1-Q4). |
| Academic | It consists of eight questions to know about the use of the platforms, the space available to attend classes, and the technological devices used (Q5, Q6, Q7,Q8, Q12, Q13, Q16, Q17). |
| Social | Three questions about virtual sessions and their interaction with teachers and classmates (Q9, Q15, Q21). |
| Emotional | Four questions to learn students' moods (Q10, Q11, Q14, Q18). |
| Economic | Three questions related to the costs of general services and expenses generated by the pandemic (Q19, Q20,Q22). |
| Health | Two questions to know the health condition and whether they have lost any relatives because of COVID-19 (Q23, Q24). |

*3.2. Participants*

The GCTPU needed to identify and evaluate the impacts COVID-19 confinement caused. In the second half of 2020, a commission of 21 teachers was formed to carry out a study and evaluate the situation and effects derived from COVID-19 on educational processes. Of the 120 TUs and 60 PUs, and with the criteria of applying the data collection instrument to institutions with the highest number of students enrolled during 2020–2021 academic period, data from 15 TUs and 7 PUs were used, guaranteeing reliability of our results [56]. Convenience sampling was used to determine the number of students who would respond to the questionnaire. This sampling is a nonprobabilistic and nonrandom technique used to create samples according to the ease of access, and availability of people to conform the sample [57], in a given time interval or any other practical specification of a particular element. It was also determined that the period of application of the survey would cover the period May–August, 2021 [56]. The instrument was answered by 6596 randomly selected students from 15 PUs and 7 TUs of Mexico.

## 4. Results

### *4.1. Analysis of Student Evaluation Instrument*

Students are the most important stakeholders in the educational system. During the pandemic, they migrated from face-to-face to virtual classes without being asked whether they agreed or not. The situation has forced them to spend several hours in front of a screen and to learn how to use new applications or platforms. This study allows us to know the situation that students live in under these conditions and allows to propose new teaching strategies for a better academic performance. The measuring instrument evaluated a total of 6596 students from the TUs and PUs at a national level. The evaluation was made regarding general, academic, social, emotional, economic, and health effects. A total of 6595 surveys were obtained. The surveys carried out were applied to students from different disciplines, including 4 from the area associated with administrative educational programs and 12 from different areas of engineering. Females participated mostly in this survey, being 51% of the participants: female participation in many fields is increasing and the educational sector is not lagging. According to the data obtained, 65% of the respondents are in the age range of 19 to 21 years. It is possible to emphasize what is shown in Figure 2 in relation to the geographical areas where some of the TUs and PUs are located. There are greater problems in those universities located in the south and southeast regions than in the universities located in the central-north region of Mexico. Table 2 presents an analysis of the most relevant responses for each category.

**Table 2.** Relevant responses.

| Effects | Responses |
|---|---|
| Academic | 39% of students do not have any concern taking their classes online. 41% of students agreed with having online classes through a platform. 17% of students surveyed do not have a space to take their classes, while 44% of students have an adapted space to take their classes. 58% of respondents do not have internet connection. 47% of students do not agree that virtual classes facilitate the teaching–learning process. 59% of students say that the best option to take classes is face-to-face. Most students believe that their teachers use the platform properly to teach their virtual classes. Teachers have received emergent training for the proper use of virtual educational. |
| Social | 55% of students perceived that there is no need for regulating coexistence between peers during their virtual classes. More than half of the students who responded to the questionnaire stated that they had participated in social gatherings more than once. |
| Emotional | 30% say they were stressed. 15% state that their family requires psychological support. This result is directly related to university students who experience stress or have lost a relative due to COVID-19. 52% of students have experienced stress. The most prevalent response among the university student community is that related to a normal state of mind, referring to a student community as calm or comfortable, or a comfortable environment that allows them to carry out pending activities, whether family or personal, considering social preventive measures to avoid contagions and probabilities of family losses due to the COVID-19 pandemic. |
| Economic | 84% of students have invested up to 10 thousand MXN purchasing technological devices to take their virtual classes. The confinement situation has generated expenses in the payment of services (internet, electricity, water, etc.). The vast majority of the student population has increased their expenses for services during the pandemic. 78% of students have had additional expenses that affect their economy. |
| Health | 64% of students have had no symptoms of COVID-19. 22% of students have suffered the loss of a family member because of COVID-19. |

*4.2. Statistical Analysis*

It was an interest of the institution to find out if there were statistical differences by gender to the effects. The questions where gender impacted the effects were the only ones considered: Q3 and Q4 were contrasted with the rest of the questions. Of the students surveyed, 51.62% were female and 48.32% were male. If the pandemic had affected males and females equally, one would expect a similar distribution on the questions related to the aversion to taking virtual classes and their status during the pandemic. In other words, the expected female-to-male ratio would be 1.07. However, the following table shows the observed female-to-male ratio for questions Q10, Q14, and Q18; see Table 3:

**Table 3.** Female-to-male ratio for emotional affectations.

|  | Q10 | Q14 | Q18 |
|---|---|---|---|
| Bored | 0.59375 | 0.5981982 | 0.7434211 |
| Happy | 0.9319899 | 0.8464286 | 1.3523316 |
| Stressed | 1.461039 | 1.4574976 | 1.3322034 |
| Cheerful | 1.2275862 | 0.7466667 | 0.9292557 |
| Sad | 1.1190476 | 1.2803738 | 1.2598425 |

Table 3 shows a higher proportion than expected for the stressed category. This indicates that females tend to feel more stressed with virtual classes with their family and about continuing with this model. The proportion of females to males who are undecided about whether or not to continue in a virtual environment is 1.19. In addition, the statistical significance of these differences between the expected proportion and the proportion obtained was evaluated by means of a Chi-square test, see Table 4, which yielded the following results:

**Table 4.** Chi-square test.

|  | Q10 | Q14 | Q18 |
|---|---|---|---|
| Statistic | 107.32 | 156.51 | 45.957 |
| *p*-value | $<2.2 \times 10^{-16}$ | $<2.2 \times 10^{-16}$ | $2.514 \times 10^{-9}$ |

It is evident from the results that the *p*-value is lower than $\alpha = 0.05$; therefore, H1 is accepted. The statistical differences happened to be less significant than in the male results which is understandable because they are less stressed as the results from Q10 and Q14 showed.

Figure 2, the left side of the graph, shows the students who are comfortable or happy with the distance learning scheme; on the right side are the students who are not comfortable with the distance learning model. In quadrants I and II, the results correspond to students who are not against it. In quadrants III and IV, students who did not feel comfortable with the online learning model are concentrated.

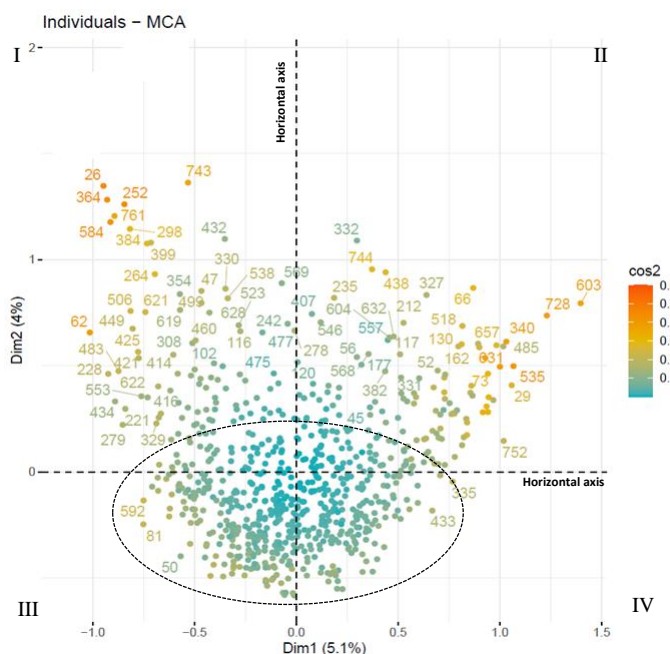

**Figure 2.** Individual Factor Map—MCA.

## 5. Discussion

A hybrid learning model refers to the blending and mixing of learning environments: the face-to-face classroom instruction and the online environment [58]. An example of this is adventure learning (AL), which is a hybrid distance education approach that provides students with opportunities to explore real-world issues through authentic learning experiences within collaborative learning environments. Problems such as infrastructure and the lack of mobile devices or connectivity have a more negative impact on the entire university community. It has been important to identify gender-related factors because the term "equity" is a complex concept to be defined: it depends on many factors, mainly political ideals [45], and this has been further demarcated with the COVID-19 crisis [46]. Lawal (2020) highlights the degree of the differences in mental health indicators of Nigerians according to sociodemographics and the importance of improving mental health during and after the COVID-19 pandemic [49]. Other studies mention that mental, physical, social, and economic impacts are attributable to pre-existing inequalities in the academy, but the affectations during the pandemic should be differentiated from the ones before the isolation conditions, such as the gender "roles" in individual family environments. Because of the culture in the country, females have more demands at home as well as being a focus in terms of the violence that can be suffered at home [59]. In the same sense, this is reflected in the greater indecision to follow a virtual model. To make matters worse, in times of stress, such as pandemics, biased decision-making processes are favored that threaten to deprioritize equity initiatives [20,38,41,50,60].

In this sense, it would be interesting to delve into the work of other researchers to determine a profile of mental health in relation to gender, which could contribute to establishing protocols to address these problems with greater specificity. According to the results of the present work, if the student is from a university located in marginalized geographical areas, the effects of anger, fear, anxiety, stress, and depression on physical, mental, and emotional health will have greater repercussions according to the results obtained by Vital (2022) [21] and the analysis related to the affectations to the TUs and PUs students in the present research. It would be interesting to find out if the students from places with scarce or no internet infrastructure really are more stressed than those in cities where the restrictions were more severe and public spaces closed compared

to students in rural areas where they can walk around or participate in the local rural economy. It has to do, to a great extent, with organic factors that immediately alert people who are at risk of the contagion. The danger of the contagion is present, and it is also closely related to massive deaths. Therefore, physiological mechanisms that alert about possible damage to life itself are set in motion but are not the same for all people as their background is different. In this sense, it would be very convenient to carry out more studies on socio-emotional factors for certain areas. It is important for university students that specific information is generated and protocols are created for the better care of the university population [61,62]. It seems that mental health, anxiety, or stress are related to the factors studied, but they are not the only ones that exist. While differentiation is a must for all teachers, it should also be a requirement for universities planning and changing strategies that behave differently according to the geographical location and gender particularities. When education is so important for developing a region, internal and external factors should be considered as the prime element for the competitive advantage of a technical education (in this specific case) over the rest of the educational institutions. According to Ng (2013) [38], unemployment is clearly a major concern in times of an economic crisis. Prospective studies unsurprisingly show that unemployment has a causal influence on depression. Common sense dictates that depression will reduce the chance of re-employment and reintegration into an already strained economy and eventually the chronically unemployed suffer increased debts. Longitudinal data show that financial difficulties lead to increased major depression, with housing payment problems and consumer debt leading to poorer mental health, in short, the quintessential "vicious cycle". Wan (2020) [39] identified the proportion of respondents showing depression, anxiety, and/or suicidal thoughts is alarming. Respondents reported academic, health, and lifestyle-related concerns caused by the pandemic. Given the unexpected length and severity of the outbreak, these concerns need to be further understood and addressed. Among 2031 participants, 48.14% showed a moderate-to-severe level of depression, 38.48% showed a moderate-to-severe level of anxiety, and 18.04% had suicidal thoughts. A majority of the participants indicated that their stress/anxiety levels had increased during the pandemic. Less than half of the participants indicated that they were able to cope adequately with the stress related to the current situation. Mann (20210) measured the impact of COVID-19 on students and noted that COVID-19 confinement really affects both the social and academic life of the student [63].

## 6. Conclusions

Through this study, the objectives were accomplished. First, we aimed to determine if there were significant differences in the physical and emotional health derived from COVID-19 in the TU and PU students. If the pandemic had affected males and females equally, one would expect a similar distribution on the questions related to the aversion to taking virtual classes and their status during the pandemic. Our research determined that beyond what affectations were significant, the negative effects were less perceived in females, although in the social affectations, females were more affected negatively. This result could lead to a differentiated teaching and learning process approach considering the extra-walls environment.

The other objective was to identify the main factors that affected the academic performance of TU and PU students during the pandemic. It was identified that the effects that were more commonly suffered were the economic ones and mostly for those students from rural areas, specifically from the southern and south-east areas. This could be explained as those being geographical zones in Mexico with less economic development, contrary to the zones in the center-north from the country, which are more economically developed. Other relevant affectations were connectivity and TICs, the lack of physical activity, and the dynamics of the class.

**Hypothesis 1 (H1).** *There are significant differences in the physical health and emotional state of the TU and PU students in relation to gender during the confinement derived from COVID-19.*

The measurement instrument applied to 6596 students from the 15 TUs and 7 PUs of Mexico showed that stress is the variable that mostly influences the results which, among the university students, include a lack of technological resources, the online model for teaching classes by teachers, family finances, and the loss of or an infected family member due to COVID-19. In the academic scope, the students reveal that internet connectivity affects their performance in virtual classes. Some also reported that they do not have an adequate space to take their classes. In addition, almost half are taking their classes using a cell phone. This can limit the development of the most elaborate works and even their delivery in a timely manner. This might make them feel that they are not learning what they should. Emotionally, half of the students have positive feelings such as joy and happiness as they are taking their classes at home with their family. In addition, students indicated that they do not require psychological support. The university hybrid model could bring great benefits in the social, economic, academic, emotional, and health fields if we consider the results found in this research in a positive way. For instance, a percentage of students express a feeling of happiness to be at home. A hybrid model would allow you to spend more time at home, strengthening family coexistence. In addition, it would allow you to do activities that you do not normally do. Healthwise, some students reported the loss of a family member due to COVID-19. Such losses could have impacted their emotional wellbeing. Also, the student population considers that there is little to almost no coexistence with their classmates. In this sense, the global engineering requirements should be published to enrich the relationships that would develop the social skills necessary for managing projects and generate innovative points of view resulting from an interdisciplinary working environment. Students mentioned that virtual meetings with friends are rare and, as such, their social lives are next to zero due to the contingency measures.

*Recommendations or Further Studies*

It is necessary to innovate and implement a hybrid education system with teaching–learning strategies considering the socioeconomic factor of the region of each country. To do this, some factors must be considered. Some of the factors may include, but not only, the selection of topics that will be taught in a physical, hybrid, or virtual mode. There must be an appropriate selection or definition of the cultural and sports activities that will complement the integral education of the student to reduce stress during their university time, allowing for their incorporation into the productive sector and the social sphere, as do students who have completed their educational plan in person. Another factor to be considered is the infrastructure that the student and the institution have, as well as the university's support services for the student. For those students who lack a study space at home or have inadequate internet access or insufficient computer equipment, among others, the university within its facilities could provide open or closed study spaces with access to the institution's internet network without necessarily being in a classroom during its virtual sessions. The students must be understood in the academic and economic dimension, so it is necessary to develop a methodology in the educational institution to identify vulnerable students who require support. Therefore, the educational institution should invest more for what could be a nonfeasible project. However, if they make class schedules grouping the subjects online so that the student does not attend the university, it will reduce the cost of water, electricity, classrooms, and infrastructure in general, among others, and could balance the operating costs of the educational institution. Research for the specific pillars of the academic world [50] should be carried out in future studies analyzing the inequity in the three academic pillars of academic assessment in the TUs and PUs: teaching, service, and research .

**Author Contributions:** Conceptualization, L.V.L. and R.G.G.; methodology, L.V.L., M.A.Z.-A., H.R.R., M.A.C.-P. and J.R.R.; formal analysis, M.A.Z.-A., W.J.P.G., M.A.C.-P. and L.V.L.; investigation, L.V.L. and F.F.R.; resources, M.C.R. and H.R.R.; data curation, W.J.P.G., M.A.C.-P. and M.A.Z.-A.; writing—original draft preparation, F.F.R. and L.V.L.; writing—review and editing, M.A.Z.-A., J.R.R., M.A.C.-P. and R.G.G.; supervision, J.R.R. and R.G.G. All authors have read and agreed to the published version of the manuscript.

**Funding:** This research received no external funding.

**Institutional Review Board Statement:** The study was conducted according to the guidelines of the Declaration of Helsinki and submitted to the participating universities for approval and distribution. The Project was approved in the Ethics Committee session. Date 2 September 2020.

**Informed Consent Statement:** Respondents consent was waived due to the minimal risk to subjects that will not adversely affect their rights and welfare. It was obtained by voluntarily answering and informing that it was for academic and statistical purposes only.

**Data Availability Statement:** Data are available on request to Lourdes Vital López.

**Acknowledgments:** The authors would like to thank to the students who responded to the data collection instrument and also, to thank to the participating universities: Universidad Tecnológica de la Selva, Universidad Tecnológica de Chihuahua, Universidad Tecnológica de La Costa Grande De Guerrero, Universidad Tecnológica de la Región Norte de Guerrero, Universidad Politécnica Metropolitana de Hidalgo, Universidad Politécnica de Huejutla, Universidad Politécnica de la Energía, Universidad Politécnica de Lázaro Cárdenas, Universidad Tecnológica Emiliano Zapata del Estado de Morelos, Universidad Tecnológica de Bahía De Banderas, Universidad Tecnológica de la Sierra Sur De Oaxaca, Universidad Tecnológica de Tecamachalco, Universidad Tecnológica de Tehuacán, Universidad Politécnica de Amozoc, Universidad Tecnológica de San Juan Del Río, Universidad Politécnica del Golfo de México, Universidad Tecnológica de Tamaulipas Norte, Universidad Tecnológica de Matamoros, Universidad Tecnológica de Nuevo Laredo, Universidad Tecnológica de Tlaxcala, Universidad Politécnica de Tlaxcala, Universidad Tecnológica del Sureste de Veracruz, and Universidad Politécnica de Zacatecas. The authors would like to thank the members of the following academic groups: the Academic Group of Chemistry of Universidad Tecnológica de San Juan del Río, the Academic Group of the Maintenance of Universidad Tecnológica de Tamaulipas Norte, and the Academic Group of Software Development of the Universidad Tecnológica de la Region Norte de Guerrero.

**Conflicts of Interest:** The population of this study are professors from TUs and PUs belonging to the General Coordinator of Technological and Polytechnic Universities (CGUTP). The data collection instrument was previously designed and then sent to the professors of each participating university with a Google Forms link. The response to the questionnaire was voluntary which made it a convenience sampling method. The survey header had a note informing about the use of the data to be collected and the voluntary response to provide answers as consent: The data will be analyzed and shared only within the research team, with identifiers destroyed. The nature of the data is electronic and does not include biological specimens or sensitive information. Only the research team had access to the data and identifiers were destroyed (anonymized data). The data were stored on a computer managed solely by the research team and password protected. The data files were managed by the research team with the identifiers already destroyed and will be kept in a file with the final version of the article as evidence of the research for institutional purposes. Respondents accessed the data collection instrument through a link sent by the Research Group, with the collected data addressed to the research team. It was an anonymous and voluntary response with minimal risk to the participants, not exposing them to psychological, social, or physical risks, belonging to the definition of research for generalized knowledge, which makes the survey of this study exempt, and it was reviewed and authorized by the Ethics Committee of each participating university.

## Appendix A

**Table A1.** Data Collection Instrument Student's opinion.

| | |
|---|---|
| **Q1. Mention your affiliation** | |
| **Q2. What academic program do you study** | |
| **Q3. Gender** | Male         Female |
| **Q4. Age Range** | 16–18 years |
| | 19–21 years |
| | 22–24 years |
| | 25–27 years |
| | 28 or more |
| **Q5. Do you agree in taking your classes online** | Strongly Agree |
| | Agree |
| | Neither agree nor disagree |
| | Disagree |
| | Strongly disagree |
| **Q6. How adequate is the space where you take your virtual classes** | Excellent |
| | Good |
| | Regular |
| | Pour |
| | Bad |
| **Q7. What are the technological devices you mostly used to take your virtual classes** | Desktop computer |
| | Laptop |
| | Cellphone |
| | Tablet |
| **Q8. Does internet connectivity speed affect your performance when taking virtual classes** | Too much |
| | More or less |
| | A little |
| | Nothing |
| **Q9. How do you consider the interaction with your peers during virtual classes** | Excellent |
| | Good |
| | Regular |
| | Pour |
| | Null |
| **Q10. How would you describe your mood being with your family while taking classes** | Happy |
| | Cheerful |
| | Sad |
| | Bored |
| | Stressed |
| **Q11. Do you feel that your family requires psychological support to continue their lives during the pandemic** | Strongly Agree |
| | Agree |
| | Neither agree nor disagree |
| | Disagree |
| | Strongly disagree |
| **Q12. How much you agree that taking virtual classes facilitates teaching and learning** | Strongly Agree |
| | Agree |
| | Neither agree nor disagree |
| | Disagree |
| | Strongly disagree |

**Table A1.** *Count.*

| | |
|---|---|
| **Q13. What option do you consider to be the best to take classes** | Take face-to-face classes |
| | Take virtual classes in a platform |
| | Some virtual ones and other ones face-to-face |
| | Practical subjects in face-to-face mode |
| | Theoretical subjects in virtual mode |
| **Q14. How have you felt taking all your classes in virtual mode (all shift)** | Happy |
| | Cheerful |
| | Sad |
| | Bored |
| | Stressed |
| **Q15. What is the mean you used to communicate with teachers** | WhatsApp |
| | E-mail |
| | Chat in the institutional platform |
| | Facebook Call |
| | much |
| **Q16. How does the lack of technological devices (computer, cell phone, tablet, laptop, Internet), influences your academic performance** | It influences too |
| | influences a lot |
| | It influences More or less |
| | It influences a little |
| | It does not influence anything |
| **Q17. Do teachers use the platform correctly to give their virtual classes** | Always |
| | Almost always |
| | Regularly |
| | Little |
| | Never |
| **Q18. Your prediminantly mood during the pandemic is** | Happy |
| | Normal |
| | Sad |
| | Depressive |
| | Stressed |
| **Q19. How much have you invested in the purchase of technological equipment to take your virtual classes** | Less than $5,000.00 |
| | From $5,000.00 to $10,000.00 |
| | than $10,000.00 to $15,000.00 |
| | than $15,000.00 to $20,000.00 |
| | More than $20,000.00 |
| **Q20. The confinement situation has generated expenses in the payment of services (Internet, electricity, water)** | Too much |
| | Regular |
| | A little |
| | Not frequent at all |
| **Q21. How often do you participate in meetings with friends through virtual media** | Very often |
| | Frequently |
| | Regularly |
| | Rarely |
| | Not at all |

**Table A1.** *Count.*

| | |
|---|---|
| **Q22. Has the pandemic situation generated additional expenses that affect your economy** | Strongly Agree |
| | Agree |
| | Neither agree nor disagree |
| | Disagree |
| | Strongly disagree |
| **Q23. Your health condition regarding COVID-19 is** | Confirmed COVID-19 |
| | Suspected COVID-19 |
| | Negative COVID-19 |
| | Symptoms related to COVID-19 |
| | None of the above |
| **Q24. Have you suffered the loss of a family member due to COVID-19** | Yes · No |
| **Q25. Mention any educational strategy (activities, exercises, etc.) that helps the teacher to improve performance in their virtual classes** | |

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
