# Peer review of "The Impacts of COVID-19 on Technological and Polytechnic University Students in Mexico"

_sustainability, doi:10.3390/su14106087_

Round 1

Reviewer 1 Report

The Impacts of COVID-19 on Technological and Polytechnic University Students

The present study aims the study aims to study the students' reaction to the change in pedagogical practice, driven by COVID. The topic is very interesting and very relevant for the institutions in México.

The abstract is well structured but is confusingly written, needs improvement. The title should clearly indicate that the study was carried out in Mexico and corresponds to the reality of Mexican society.

The text is poorly worked, the same ideas are repeated throughout the text. It should be revised and more summarized. For example: “The questions asked were related to the social, economic, academic, emotional, and health effects experienced. The evaluation tool focused on the social, economic, academic, health and emotional wellbeing of the students”.

The objectives of the study are clearly and explicitly defined at the end of the introduction. No need to repeat at point 1.3.

The introduction confuses the use of mobile devices in face-to-face and distance learning. The discussion of advantages and disadvantages has to be differentiated. The framework repeats the lack of rigour in the discussion. The use of teaching support tools, such as social networks, have different effects if used in face-to-face or distance learning.

The framework writing should be boiled down to what is fundamental to the article. Pedagogical phrases like " Anxiety can be considered as a normal adaptation response to what is considered a threat (stress). Since it allows a person to improve their performance, even when sometimes this response may not be adequate. Stress can be excessive for the resources available to the individual though." should be avoided because they disperse the reader's attention.

The reference to the Preliminary study is not well justified. It does not seem to add anything to the article and could be interpreted as self-citation.

Hypothesis 1 and hypothesis 2 are the same.

The need to include the Problem Statement point is not understood. Repeats what was written in the introduction.

The introduction of the methodology should be significantly reduced and figure 1 eliminated. It is appropriate for a master's or doctoral thesis but not for an article.

It is not clear whether the questionnaire has 24 or 25 questions.

Some questions do not seem to be formulated clearly for the respondents. For example: “How much do you agree that you have to take your classes through a platform.” During the pandemic it was mandatory to use digital platforms, students had no choice.

Another example is the question “How have you felt in this contingency being at home with your family”. Surely at different times they experienced all the sensations asked about. Was the question multiple choice?

The characterisation of the sample is in a very confusing format and difficult to understand. It should be transformed into a table. For example, the point 4.2. General effects, is not understandable.

The same comment for the results of the questions asked. It must be possible to read the results clearly. It is proposed to rewrite them in table format.

The results should be compared with the results of similar studies and the study has limitations which have not been discussed.

Overall, the text needs to be revised to improve its readability. The results should be presented in tables and the level of detail in non-essential information should be avoided. The questions put to the students need to be better explained. Some of them seem to be badly formulated.

Author Response

Reviewer 1

The present study aims the study aims to study the students' reaction to the change in pedagogical practice, driven by COVID. The topic is very interesting and very relevant for the institutions in México.

The abstract is well structured but is confusingly written, needs improvement. The title should clearly indicate that the study was carried out in Mexico and corresponds to the reality of Mexican society.

R=Thank you very much for your comments, the abstract has been improved see lines 1-19

The text is poorly worked, the same ideas are repeated throughout the text. It should be revised and more summarized. For example: “The questions asked were related to the social, economic, academic, emotional, and health effects experienced. The evaluation tool focused on the social, economic, academic, health and emotional wellbeing of the students”.

R= Thank you very much for your suggestions, changes and adjustments have been made.

The objectives of the study are clearly and explicitly defined at the end of the introduction. No need to repeat at point 1.3.

R= Thanks for the suggestion. The adjustment was made in the introduction and specific objectives clarified. See lines 232-239

The introduction confuses the use of mobile devices in face-to-face and distance learning. The discussion of advantages and disadvantages has to be differentiated. The framework repeats the lack of rigor in the discussion. The use of teaching support tools, such as social networks, have different effects if used in face-to-face or distance learning.

R=Thanks for the suggestion, the introduction part was improved. See lines 38-140

The framework writing should be boiled down to what is fundamental to the article. Pedagogical phrases like " Anxiety can be considered as a normal adaptation response to what is considered a threat (stress). Since it allows a person to improve their performance, even when sometimes this response may not be adequate. Stress can be excessive for the resources available to the individual though." should be avoided because they disperse the reader's attention.

R=Thanks for the comment, improvements have been made See lines 141-186

The reference to the Preliminary study is not well justified. It does not seem to add anything to the article and could be interpreted as self-citation.

R=Thanks for the comment, removed the phrase and the reference.

Hypothesis 1 and hypothesis 2 are the same.

R= Thanks for the observation, H1 corresponds to the alternative hypothesis and H2 corresponds to the null hypothesis. See lines 241-246

The need to include the Problem Statement point is not understood. Repeats what was written in the introduction.

R= It has been included in this way in order to have a clearer understanding of the problem, but it was edited to avoid repetition. See lines 247-262

The introduction of the methodology should be significantly reduced and figure 1 eliminated. It is appropriate for a master's or doctoral thesis but not for an article.

R=This has been included to provide clarity to the reader on the development of the steps to carry out the research work.

It is not clear whether the questionnaire has 24 or 25 questions.

R=There are 25 questions in the data collection instrument. See lines 288-290

Some questions do not seem to be formulated clearly for the respondents. For example: “How much do you agree that you have to take your classes through a platform.” During the pandemic it was mandatory to use digital platforms, students had no choice.

Another example is the question “How have you felt in this contingency being at home with your family”. Surely at different times they experienced all the sensations asked about. Was the question multiple choice?

R= Thank you very much for your comment, the adjustments have been made. See Appendix A

The characterization of the sample is in a very confusing format and difficult to understand. It should be transformed into a table. For example, the point 4.2. General effects, is not understandable.

The same comment for the results of the questions asked. It must be possible to read the results clearly. It is proposed to rewrite them in table format.

R= Thank you very much for your comment, the adjustments have been made. See Table 2.

The results should be compared with the results of similar studies and the study has limitations which have not been discussed.

R= Thank you very much for your comment, improvements have been made.

Overall, the text needs to be revised to improve its readability. The results should be presented in tables and the level of detail in non-essential information should be avoided. The questions put to the students need to be better explained. Some of them seem to be badly formulated.

R=Thank you very much for your comment, improvements have been made.

Reviewer 2 Report

Dear authors

I would like to congratulate you on your article and express my admiration for your efforts and endeavors in relation to the manuscript "The Impacts of COVID-19 on Technological and Polytechnic University Students".  

I appreciate the structure of the study as well as your research and its processing.
I also appreciate the structure of the system, which is described in Figure 1.

A certain shortcoming of the study is narrow publication focus on studies in Spanish, and therefore I suggest you to add studies of several authors, who´s work will expand (by reference) your study with other assumptions and observations that deal with the issue.

For example: 

Tkacova et. al. https://doi.org/10.3390/su131810442  (Central Europe) Tkacova et al. https://www.mdpi.com/1660-4601/19/5/2767 (Central Europe)  Pavlikova et. al. https://doi.org/10.3390/su131810350  (Central Europe) Petrovic et al.  https://doi.org/10.3390/su131910826  (Czechia) Ružić-Baf  et al. https://doi.org/10.15503/jecs2021.2.399.411  (Croatia)  Kobylarek et al. https://doi.org/10.15503/jecs2021.2.5.11  (Poland) Pooja Mann  et al. https://doi.org/10.15503/jecs2021.2.361.374  (India)   I recommend publishing the text after completion. 

Author Response

I would like to congratulate you on your article and express my admiration for your efforts and endeavors in relation to the manuscript "The Impacts of COVID-19 on Technological and Polytechnic University Students".  

I appreciate the structure of the study as well as your research and its processing.
I also appreciate the structure of the system, which is described in Figure 1.

A certain shortcoming of the study is narrow publication focus on studies in Spanish, and therefore I suggest you to add studies of several authors, who´s work will expand (by reference) your study with other assumptions and observations that deal with the issue.

R=Thank you very much for your suggestions.  The references have been included

Round 2

Reviewer 1 Report

The authors have introduced most of the suggested improvements in the paper.